# Application of GNSS/INS and an Optical Sensor for Determining Airplane Takeoff and Landing Performance on a Grassy Airfield [note 1]

**DOI:** 10.3390/s19245492

**Published:** 2019-12-12

**Authors:** Jaroslaw Pytka, Piotr Budzyński, Jerzy Józwik, Joanna Michałowska, Arkadiusz Tofil, Tomasz Łyszczyk, Dariusz Błażejczak

**Affiliations:** 1Faculty of Mechanical Engineering, Lublin University of Technology, 20-618 Lublin, Poland; p.budzynski@pollub.pl (P.B.); j.jozwik@pollub.pl (J.J.); atofil@pwsz.chelm.pl (A.T.); tomasz.lyszczyk@gmail.com (T.Ł.); 2The State School of Higher Education, The Institute of Technical Sciences and Aviation, 22-100 Chełm, Poland; jmichalowska@pwsz.chelm.pl; 3Department of Construction and Usage of Technical Devices, West Pomeranian University of Technology in Szczecin, 70-310 Szczecin, Poland; Dariusz.Blazejczak@zut.edu.pl

**Keywords:** airfield performance, landing and takeoff distances, rolling resistance coefficient, GNSS/INS sensor, optical sensor, measurement, grassy airfield

## Abstract

The performance of a PZL 104 Wilga 35A airplane was determined and analyzed in this work. Takeoff and landing distances were determined by means of two different methods: one which utilized a Global Navigation Satellite System/Inertial Navigation System (GNSS/INS) sensor and another in which airplane ground speed was measured with the use of an optical non-contact sensor. Based on the airfield measurements, takeoff and landing distances as well as rolling resistance coefficients were determined for the used airplane on a grassy runway at the Radawiec airfield, located near Lublin, southeast Poland. The study was part of the “GARFIELD” project that is expected to deliver an online information system on grassy airfield conditions. It was concluded that both sensors were suitable for the aimed research. The results obtained in this study showed the effects of high grass upon the takeoff and landing performances of the test airplane. Also, the two methods were compared against each other, and the final results were compared to calculations of ground distances by means of the chosen analytical models.

## 1. Introduction

### 1.1. Grassy Airfields

General aviation (GA) airplanes often operate on aerodromes with unprepared, grassy runways. Some examples of such operations are agricultural flights, firefighting, humanitarian flights in the third world countries, search and rescue flights, and sport and leisure aviation. Airfield performance, in terms of ground roll distances, is strongly affected by weather conditions. In order to ensure safe and efficient takeoffs and landing, it is necessary to analyze the performance of an airplane, and this is mainly affected by wheel–soil interactions. This study is an extended version of the paper entitled “Measurement of Takeoff and Landing Ground Roll of Airplane on Grassy Runway”, presented during the 2019 IEEE International Workshop Metrology for Aerospace in Torino, Italy [1]. 

Grassy runways are one example of a natural terrain, and its mechanical strength is very sensitive to changes in moisture content [2,3,4]. A dry grassy surface is hard, and a wheel does not sink down. Wetting the grassy surface results in a decrease of strength which may have bad effects on airfield performance [5].

The two most important issues are (1) a decrease in bearing capacity which results in a reduction of gross weight and (2) an increase in the rolling resistance of the wheels. A solution by means of an instrumental method for the first problem was the introduction of the California Bearing Ratio (CBR) which is a measure describing mechanical characterization of homogenous soil. Four different categories of flexible surfaces were introduced by the International Civil Aviation Organization (ICAO): high, medium, low, and very low. These measures were quantified by means of the CBR values 15, 10, 6, and 3, respectively [6]. 

The problem of increased rolling resistance is critical for safe and economical operation of aircrafts on grassy airfields. Rolling resistance, *F_RR_*, determines the dynamics of the longitudinal motion of wheeled vehicles including aircrafts. Generally, the higher the *F_RR_*, the longer the takeoff distance (landing distance decreases, but the problem of low CBR may result in a nose-down incident). The rolling resistance is often mixed-up with braking friction. This is false, since the braking friction is, in fact, the coefficient of friction between the wheels and a surface. Besides, knowing the actual rolling resistance is critical in the determination of the airfield performance of an airplane [7,8,9,10].

The aim of this study was to measure the airfield performance of a light GA airplane operating on a grassy airfield under mixed surface conditions. A comparison of two sensor systems was also conducted. In addition, we compared analytical models that enable the calculation of the takeoff and landing ground distances.

The difficulty of measuring airfield performances of an airplane during takeoff or landing on grassy airfields lies mainly in the fact that the tire–soil interactions on a deformable grass surface depends on many factors such as soil moisture, grass length, humidity, and the weather. The high dynamics of changes in these factors means that traditional models of forecasting the aircraft’s ground performance give vague results, differing from the real ones. On the other hand, measurements in real conditions using full-size objects (aircraft) are burdened with errors whose impact is difficult to estimate. Hence, there is importance in this undertaken research for the larger research community and for practitioners in general aviation. The subject matter is rarely discussed in the literature, and there are no proven and credible solutions. The cited methods of testing the takeoff and landing distances of an airplane are either not very accurate or complicated and difficult to access, especially in the case of grassy airfields. Therefore, the search for a new method or adaptation of existing solutions may contribute to a more rational and effective use of grassy airports, improvements in safety, and extension of the network of connections in air communication.

### 1.2. Methods for Determination of Rolling Resistance for Aircraft Tires: A Review

A revision of methods for rolling resistance determination was done by van Es [7] who developed a method for the determination of an aircraft’s wheel rolling resistance in snow as well as by Shoop [9] who determined a method to estimate the rolling resistance of an airfield to predict takeoff distance.

Van Es assumed that high-speed compaction of a compressible material (e.g., soil, snow, grass) causes a resistance related to the speed with which the material is compressed. It was suggested that this resistance is related to the increase in kinetic energy of the particles from zero up to the aircraft ground speed. The resulting method assumes two terms: drag due to the compression and drag due to the motion. This method has limitations, one of them was that it is not recommended for wet snow or wet soil [7].

Shoop concluded that the total rolling resistance of an aircraft on loose soil is due to the internal mechanical resistance (e.g., landing gears, wheels suspensions, bearings), aircraft aerodynamic drag, low-speed compaction of the soil, and high-speed resistance due to the soil drag on the wheels and spray drag on the body of the aircraft. Soil compaction term was determined with an instrumented vehicle, and the speed-related term was estimated in real size flight tests in which a C-17 transport aircraft was used [9].

Rowe and Hegedus [10] have dealt with rolling resistance on extremely wet soils and proposed an equation in which the resistance of a wheel is due to the viscosity and static pressure and to velocity pressure (dynamic resistance). 

Crenshaw [11] developed an equation for rolling resistance due to the fact of velocity and dynamic sinkage which is a function of tire diameter, deflection, section height, cone index, and other empirical constants. These factors can be determined from soil data and from measurements on an instrumented aircraft.

Hovland [12] developed a theory of inertial forces in moving soil and their effect on tractive forces. It was possible to predict at what velocity severe and immobilizing sinkage of the wheels will begin for an airplane landing on a soft surface. As an example, a theoretical relationship between soil lift force and aircraft ground speed for a Cessna 150 landing on soft playa or marsh was shown.

Gibbesch [13] presented a simulation method for calculation of tractive forces on wheels of an aircraft on soft soil. This method assumes the effect of both elasticity and viscosity on contact pressures in wheel–soil systems, and the soils were described by means of a rheological model. The method was applied to determine the vertical forces acting on the multi-wheeled gears of a transport aircraft.

### 1.3. Takeoff and Landing Ground Roll Distance Determination Methods

Takeoff and landing belong to the most critical maneuvers of airplanes, especially when operated on unpaved, short grassy runways. From the point of view of a pilot, a safe takeoff ground roll must be finished well before the runway ends, mainly because of obstacles clearance. It is a similar situation for landing, where the approach should take into account a safe clearance above obstacles. This situation is shown in Figure 1. A pilot preparing for takeoff or planning a flight must know the ground performance of her/his airplane related to the available runway length. Because of environmental factors (e.g., grass length, soil moisture content, wind component, density altitude), an actual takeoff or landing ground roll for a given airplane may differ very much. This is a major reason why takeoff and landing performance should be measured and analyzed.

Several measurement methods for airplane ground roll during takeoff or landing have been developed through the years [14]. Some of them are briefly reported below.

The sighting bar method uses one or two sighting bars located at a known distance from the runway. By the use of these bars, a stopwatch, runway observers, and hand-recorded flight data, takeoff and landing distances data may be obtained. In the triangulation method, a camera and a scale located near the camera, parallel to the airfield direction, are used. The scale has two wires installed so that the gap between the scale and the wires (“runway wire” and “50 ft wire”) is set to coincide with the runway centre line and the 50 ft (or 15 m) screen at the mid-point of the runway. The device is set level and parallel with the runway.

The movie theodolite method utilizes a movie camera that is used to record other data, such as time and azimuth, in addition to filming the airplane. In onboard theodolite method, a camera is mounted on the airplane to obtain three-axis position information. Runway lights and other objects on or along the runway of known size are used with photogrammetric techniques and perspective geometry to obtain airplane position and altitude.

Methods based on electronics include Del Notre Transponder, laser altimeter, and global positioning system (GPS) sensors. The Del Notre Transponder measures horizontal distance and, when combined with a radio altimeter for height information, provides all the distance and height information necessary for determining takeoff and landing distances. The Trisponder consists of a distance measuring unit (DMU), a master transponder, and associated antennae, cables, and power sources.

The laser method uses a laser landing altimeter. This unit measures distance using a modulated laser beam with centimeter accuracy.

A GPS sensor is used to determine aircraft longitudinal and vertical position together with time coordinates during a takeoff or landing. The weakness of this method is that the moment of liftoff is difficult to be determined with required accuracy.

A wheel dynamometer, also used in the automotive, can be either a rotating sensor, embedded in a roadwheel or a stator sensor, in the wheel hub. A critical point is bearing the wheel, since the relatively large size of the stator sensor usually leads to unconventional solutions and major changes in wheel rim construction. On the other hand, with rotating dynamometers one has to consider the need of transferring data signals from the rotating sensor which can be done by means of a slip ring or a radio telemetry system. Also, a disadvantage of rotating wheel dynamometers is that four of the six components measured are dependent on the wheel position (the sensor rotates and changes its orientation related to acting forces). This requires recalculation of the measured data with respect to the wheel position which has to be determined independently. For our dynamometer system, we chose the stator sensor as a simpler solution. Since the space available for the sensing element core and the transducers was very tight (approximately 150 mm in diameter × 200 mm width), it was impossible to fit all the needed transducers into the stator sensor, and the *M_Y_* sensor was placed off the stator sensor and embedded into the brake arm (see the following section).

## 2. Materials and Methods

### 2.1. GNSS/INS Sensor

The GNSS/INS (global navigation satellite system/inertial navigation system) is a system based on GPS, upgraded by adding a ground reference station that is wirelessly connected with the vehicle in motion and improves the performance of the basic system. The referencing system recalculates the outgoing data and, being stationary, assures higher accuracy of the readings. The inertial navigation system (INS) consists of a set of laser fiber gyroscopes and accelerometers that enable the calculation of the kinematics of a running (flying) vehicle, based on the measured linear and angular accelerations. The results of those measurements, in the form of time courses, are then integrated in order to obtain velocities (1st integral after time) and distances (2nd integral after time). 

The system used in this study was the OXFORD RT3002 (Oxford Technical Solutions Ltd., Oxfordshire, UK) which consisted of a main unit installed in the aircraft with a 12 V power supply, a remote base station, and a portable computer for data acquisition and storage. The system had to be initiated before tests by moving with a constant velocity of approximately 10 km/h, and it was performed in a ground vehicle. Initiating the system in the airplane was not possible because of the high vibration level. The acquisition time of the system was 10 ms, and the positioning resolution was 20 mm. The navigation unit enabled measurement of the 39 kinematical parameters of the aircraft’s motion. Figure 2 shows the installation of the GNSS/INS unit in the test airplane.

### 2.2. The CORREVIT L400 Optical Sensor

The CORREVIT optical sensor is a type of sensor used typically in the automotive industry or in research. It consists of a light emitter that illuminates the surface, and the scattered light is analyzed in the optical receiver. Since the motion of an object (i.e., vehicle, airplane) causes changes in the light wavelength, relative velocity can be determined with accuracy of 0.1 km/h and the range of measurements is 0–250 km/h. The main subsystem was the light emitter/receiver which was installed on a vehicle so that the distance between the emitting lamp and the road or airfield surface was kept in a required range. Changing this distance, for example, due to the fact of vehicle hopping, results in missing data. Therefore, for this study, we used a special version of the CORREVIT sensor, the L400 (Kistler Group, Winterthur, Switzerland), which features a high sensor with a surface base distance of 400 mm as well as an extended range of tolerant changes in this distance: ±130 mm. This is of critical importance when the sensor has to be used on an airplane operation on a grassy airfield. Another issue was the installation of the sensor on the test airplane. The emitting/receiving unit had to be installed outside the vehicle with a close proximity to the ground surface, but since common installation racks are for road vehicles, we had to construct an untypical installation system. This system, shown in Figure 3, ensured safe mounting of the sensor. The complete CORREVIT system consisted of additional electronics and a 12 V power supply that were placed in the airplane together with a SONY DAT digital tape data recorder (Sony Corporation, Minato, Tokyo, Japan) which was used for saving the gathered data. The use of the DAT recorder instead of a Notebook computer was due to the much higher reliability of data recording.

### 2.3. The Test Airplane

A four seat, single engine, multipurpose STOL (short takeoff and landing) aircraft PZL 104 Wilga (PZL Warszawa Okęcie, Warszawa, Poland) was used for the tests (see Figure 4). The airplane is a high wing monoplane, powered by the AI 14R 192 kW radial 7 cylinder engine (PZL Kalisz, Kalisz, Poland). It has a non-retractable, tail-dragger-type landing gear with main wheels with brakes, and low-pressure tires of 500 × 200 mm in size. The main legs are rocker type with oleo-pneumatic shock absorbers. They are castored and have a positive rake angle of 18°; the axle offset is 400 mm. The airplane is 8.10 m in length with a wing span of 11.2 m and wing area of 15.5 m^2^. The empty mass is 900 kg. In the flight tests, there were four persons on board and the takeoff weight with 125 liters of fuel was 1150 kg.

### 2.4. Airfield Conditions and Procedures

The measuring systems were maintained by two persons on board. The tests were conducted on a sport airfield in Radawiec, near Lublin, Poland. Tests on low (10 cm) and mid-high (20 cm) grasses were conducted. For both grass conditions, a total of 10 flights were performed, 5 of them with flaps in takeoff position and 5 with no flaps during takeoff. On landings, flaps were extended to normal landing position for all 10 flights.

### 2.5. Data Reduction Methods

Data from the GNSS/INS system contained numerous kinematics measures of aircraft motion, but only a few of them were used and evaluated: (1) height (altitude) above the ground, (2) ground speed, and (3) takeoff or landing ground roll. Of high importance for the precision of the analysis was to determine the time moment when the aircraft lifts off and when it touches down. This was done on the altitude graphs. Example altitude graphs are shown in Figure 5. Here, we had the time points of the lift off and touch down. Knowing the time coordinates, it was possible to analyze the mentioned measures (i.e., altitude, speed, and ground roll), and typical sets of graphs for takeoff and landing are shown in Figure 5. The numerical data were collected for further analysis (integrating the velocity courses).

The data from the optical sensor was analyzed regarding two additional inputs:

Manually given by a test engineer who was observing the undercarriage wheel of the test airplane. This input was given by simply clicking the microphone, connected to the DAT recorder, and marking the time point when the airplane lifted off;Instrumentally obtained by analyzing the outgoing signal when it vanished due to increasing altitude.

## 3. Results

### 3.1. Airplane Ground Speed Profiles

These results are from the optical sensor. Figure 6 shows two sample airplane ground speed courses versus time. A comparison can be made between the two airfield conditions: short, wet grass and long, dry grass. We can see that the difference was in the early stage of the takeoff, when the airplane accelerated intensively. The difference was in the course of the ground speed: for the short, wet grass, this course had a less intensive increase in the first few seconds. This difference was quite surprising. It may be caused by a greater rolling resistance from the uneven pavement which is more pronounced when the grass is low. Long grass compensates for surface irregularities, and then the increase in speed during takeoff is more intensive.

Based on the determined ground speed profiles, it was possible to calculate the rolling resistance coefficient. In order to determine rolling resistance, the captured speed data were analyzed by solving a differential equation of the airplane’s horizontal motion as below:(1)$$\frac{dV}{dt}=\frac{g}{W}\left[T-D-k_{RR}\left(W-L\right)\right]$$
where *V* = aircraft ground speed, *W* = aircraft weight, *T* = thrust, *D* = aerodynamic drag, *L* = aerodynamic lift, *g* = gravity acceleration, 9.81 m/s^2^, and *k_RR_* = rolling resistance coefficient.

This equation includes both aerodynamic forces, lift and drag, as well as the propeller thrust. The aerodynamic forces were calculated with the classic equations. The aerodynamic coefficients were determined from experimental data based on the assumptions:in the after-liftoff flare, the aerodynamic lift force is equal to aircraft’s weight with respect to climb angle:
(2)$$L=\frac{1}{2}\rho V^{2}SC_{L}=Q\cos\beta$$the aerodynamic drag is equal to the propeller thrust force minus the longitudinal acceleration of the entire aircraft (determined from speed data for the after-liftoff flare):
(3)$$D=\frac{1}{2}\rho V^{2}SC_{D}=T-\frac{Q}{g}a_{x}$$
where *C_D_*, *C_L_* = aerodynamic coefficients, *S* = reference area, and *ρ* = air density.

Aerodynamic data for the airplane were provided by the producer, but in order to determine true engine–propeller thrust, we performed a test in which the thrust was measured with the use of a load cell at takeoff RPM settings of the airplane engine. 

Having all parameters values, we performed data integration. Using the Euler method, the takeoff speed course was divided into several small-time intervals (i.e., 10), assuming the acceleration was constant during each interval, and the rolling resistance coefficient *k_RR_* was obtained for the end of each time interval (Figure 7). The complete algorithm for determining the *k_RR_* from the speed courses is included in the work by Pytka [8]. It is worth noting that the value of rolling resistance coefficient is not constant in the takeoff speed range, and, additionally, the course of this coefficient value decreases with the area of momentary increase in the middle range of the takeoff speed. The difference between the two surfaces—short grass and long grass—was significant, although the nature of the course remained similar. The shape of the course was interesting and may result from rheological effects occurring in plant matter (in the grass). Namely, the effect of deformation speed caused an increase in the elasticity of the grass material which was manifested by lower losses associated with the absorption of deformation energy. It happened at higher speeds, and, at low speed, we observed the opposite. In the middle range, on the other hand, the *k_RR_* value was influenced by aerodynamic forces that relieved the chassis wheels.

### 3.2. Distances of Takeoff and Landing of the Wilga Airplane on a Grassy Runway

During the first experiment, there was a 5 m/s wind and the aircraft was flown facing the wind. The results of the takeoff and landing distances are presented graphically in Figure 8. There averaged values of distance with error bars are shown.

In this experiment, the aircraft was taking off with normally deployed flaps (21°). On the second day, the wind was 0–1 m/s, and the results were much different. We performed tests with deployed flaps as well as with no flaps. The averaged values are presented together with error bars in Figure 9. The takeoff and landing distances were significantly longer. The effect of grass length was the probable cause, although we were unable to analyze the effect of wind precisely. Besides, based on the comments by the pilots, this airplane was not as sensitive to headwinds during takeoffs; this is the effect of a powerful engine with shaft reduction and high diameter propeller.

Averaged values of takeoff and touchdown speed (*V_TO_* and *V_TD_*) for the two modes of operation (flaps and no flaps) are presented graphically together with error bars in Figure 10. Note that the values for the no-flaps mode were higher; moreover, in this mode, the difference between *V_TO_* and *V_TD_* was higher. We tried the “clean configuration”, inspired by one piloting handbook which suggested it as a remedy for shortening the takeoff distance. This suggestion is probably right for a hard-paved runway, but, based on the results of our flight tests, not for the natural, soft surfaces of a grassy airfield. The touchdown speed was almost identical for the two configurations—flaps and no-flaps; this is because the landing was always performed with extended flaps (the landing configuration for this aircraft was 44° flaps). 

## 4. Comparison of the Two Sensors

Two sensors were enabled to measure data that was required to determine the ground roll distances of airplane takeoff and landing. However, the methods differed from each other in some details. It will be discussed below.

### 4.1. Accuracy and Functionality

One of the most important features of a measuring method is its accuracy. The accuracy of the method depends generally on two important factors:

Accuracy of measurement of the basic parameter;Human factor—manual determination of the lift off or touchdown.

The accuracy of the measurement of basic parameters can be determined by means of known methods or can be read from a measuring device’s data sheet. In the case of the optical sensor, it is known that its accuracy within the whole measuring range (0–250 km/h) is 0.01 km/h. From that, we could calculate the optical sensor’s accuracy of distance determination. *A^OM^*, taking into account the data sampling rate of 1 ms:*A^OS^* = (0.01 km/h / 3.6) × 0.001 s = 0.0027 m(4)

On the other hand, the GNSS/INS method is based on acceleration measurement, and its accuracy is 0.01 m/s^2^. The acquisition time was 10 ms. The GNSS/INS method’s accuracy of distance determination *A^DGPS^* is as below:*A^GNSS^* = (0.001 m/s^2^ / 0.01) × 0.01 s = 0.001 m(5)

Here, we should point out that the acquisition time could be set for 1 ms and, thus, the accuracy would be one order higher, but it was impractical, since the data files would be enormous (approximately 100–150 MB for one flight test). Besides, the accuracy of the both measuring methods was high enough. For a reliable verification of the GARFIELD models, an accuracy of approximately 1 m was enough.

However, the second factor affecting the total accuracy of the methods may be of critical importance. This was because of the manual determination of the liftoff or touchdown moment. Simply, the test engineer observed the landing gear wheel during the takeoff or landing and sent an impulse to the data recorder. This factor for the optical sensor method can be set for 1 s, based on a general human’s time of reaction. So, the human factor for the optical method is as follow:*HF^OS^* = 1 s × 27 m/s = 27 m(6)

On the other hand, this factor for the GNSS/INS sensor depends upon the method of data reduction which is described in the second section of this paper. When analyzing the time courses of the acceleration, it is noticeable that the amplitude of acceleration suddenly drops—this is the moment when the airplane lifts off. But the exact time point is difficult to determine. A method of simple discrimination has been applied, and it was assumed that the liftoff point is where the amplitude reaches 50% of its value, taken from the ground roll reading. A time between the 50% and low amplitude is set as a determinant for the accuracy (Figure 11). Therefore, the human factor for the GNSS/INS method is as follow:*HF^GNSS^* = *t^50%^* × 27 m/s = 5.4 m(7)
where a typical value of the *t^50%^* = approximately 0.1 s.

From the above, it can be seen that human factors may lower the accuracy of the methods significantly. Both human factors *HF^OS^* and *HF^GNSS^* can make the methods’ accuracies unacceptable for GARFIELD modeling purposes.

### 4.2. Installation and Operation

The installation of the CORREVIT optical sensor was difficult. The light emitter/receiver unit was installed outside the cabin at a certain position above the ground. This required some improvisation by constructing the fixture. Another issue was that the unit was left outside for the flight test, exposed to possible collisions with things inducted by the propeller. The operation, on the other hand, was very simple, since the sensor sends linear signal and we used a DAT recorder which was very comfortable for the onboard test engineer. 

The GNSS/INS sensor required to be installed by fixing in any location in the cabin. Its operation was rather difficult, since the measurement was controlled from a PC notebook computer which had to be on during the whole test flight. The onboard test engineer had to operate data acquisition software in a vibrating cabin, and there was one case when the data was lost due to the problems with operation of the computer during the flight.

### 4.3. Cost and Other Issues

Both sensor systems are expensive, although the GNSS/INS sensor is significantly more expensive compared to the CORREVIT optical sensor. The CORREVIT sensor that we used belongs to the Lublin University of Technology, where it is used for automotive applications; thus, we had uses it at zero cost. On the other hand, the GNSS/INS sensor that we used belonged to one military institute, and we had to pay for renting it for our tests. When analyzing the costs, it is important that, for both sensors, a professional should also be hired to operate it.

Table 1 includes a summary of the sensors comparison. The authors aimed to develop a measuring system based on a low-cost GNSS receiver. Such a system should have enough accuracy for very short baselines, such as the takeoff and landing distances, analyzed in this work [15].

## 5. Verification of Measurements with Calculated Values of Takeoff Ground Roll Distances

The next step in this research was to compare the obtained experimental results of the takeoff ground roll distances with the calculated results. The literature describes several equations with different degrees of simplifications and assumptions that allow for a calculation of takeoff and landing distance [16,17]. We chose three different approaches: a simple equation used by designers of homebuilt aircrafts, a general equation for airfield performance, and an equation for GA aircrafts for design projects.

### 5.1. Homebuilt Designer Equation

This calculation method was presented by Jarząb [18] and was proposed for amateur builders and designers of airplanes. It is important that homebuilt aircrafts are mostly designed to operate on grassy runways with poor surfaces. Their design utilizes tubes and fabric structures for fuselages and wooden wings with fabric covering. The overall aerodynamics of such airplanes can be poorer than factory-built, so the effective takeoff performances are critical factor. The proposed equation can be used for calculations of preliminary ground performance during the advanced stages of a design as well as in the flight-testing process. It consists of two equations: one for determining the rate of climb and the second one for calculating the final ground roll. The rate of climb equation is:(8)$$w=\frac{57\frac{R^{2}}{S}}{{\left(\frac{Q}{N}\right)}^{2}+\frac{Q}{S}}$$
where *w*—rate of climb, *R*—wingspan, *S*—wing surface, *Q*—aircraft mass, *N*—engine power.

The final ground roll equation is:(9)$$L=K\frac{{\left(\frac{Q}{S}\right)}^{2}}{w^{2}}$$
where *L*—ground roll and *K*—an empirical coefficient.

This approach is very simple and assumes the airplane is powered with a low HP engine and typical landing gear with low pressure tires. No environmental impact is regarded in the equation by Jarząb, so the effect of grass or soil deformability is neglected. The *K*-coefficient could probably be used to compensate for those effects; however, no further information is provided in the cited publication. 

### 5.2. Filippone Equation 

The second approach chosen for our comparison was presented by Filippone [19] and assumes the variable rolling resistance coefficient, *k_RR_*, which can be either measured experimentally or, for the purposes of this analysis, taken from the literature. However, the value of *k_RR_* is assumed to be constant which is in opposition to the results obtained by the present authors (Figure 7). This approach can be useful when applied to takeoffs and landings on grassy airfields with unpaved runways. The following is the final equation for ground roll distance:(10)$$S_{G}=\frac{mV^{2}}{2\frac{\eta P}{V}-\frac{1}{4}\rho A⥂C_{D}V^{2}-f_{t}mg}$$
where *m*—aircraft mass, *V*—liftoff speed, *η*—propeller efficiency, *P—*engine power, *A*—wing surface, *C_D_—*aerodynamic drag coefficient, *ρ*—air density, *f_t_*—rolling resistance coefficient, and *g*—gravitational acceleration.

For the calculations we assumed a propeller efficiency of 0.5, takeoff speed of 27 m/s, drag coefficient of 0.04, and standard air density. Other parameters were obtained experimentally, for example, the aircraft was weighted with 4 persons on board, and the engine/propeller thrust was measured. Details are included in the publication by Pytka [8].

### 5.3. Rapid Takeoff Distance Equation

The third calculation method was defined by Gudmundsson [20]. This approach is the most general and requires detailed knowledge about aircraft parameters, coefficient values, and weather conditions during takeoff. This method requires calculating the acceleration of the aircraft first and then the final ground roll distance which is very similar to the approach by Raymer [16]. The basis equation can be adopted to both turbine and piston engine powered, typical GA aircrafts of a total mass below 5700 kg. However, this approach can be simplified to one equation when applied to piston-powered light airplanes:(11)$$S_{G}=\frac{V^{2}W}{\frac{\eta_{P}\left(550P\right)}{V}g+\frac{1}{2}g\rho V^{2}S\left(\mu C_{L}-C_{D}\right)-2g\mu W}$$
where *V*—speed at liftoff, *W*—aircraft weight, *η_p_*—propeller efficiency, *P*—engine power, *S*—wing surface, *g—*gravitational acceleration, *ρ*—air density, *C_L_*—aerodynamic lift coefficient at takeoff, *C_D_*—aerodynamic drag coefficient at takeoff, and *µ*—ground friction coefficient.

### 5.4. Results Comparison

Figure 12 shows a simple comparison of all calculated and measured results. The esults obtained from both an amateur constructors’ method (211 m) and the Filippone equation (213 m) were surprisingly close to the measured average (234 m). Theoretical ground roll distance calculated using the Gundmundsson equation (304 m) was further from the experimental results. This substantial difference is suspected to be caused by the nature of the Gundmundsson equation which is suitable rather for faster GA airplanes, turboprops or high HP piston machines that operate from rigid surfaces. Nevertheless, even the simplest of the compared methods proved itself to be effective when tested against real measured values.

## 6. Conclusions

Flight test experiments on airfield performance for takeoff and landing of a light airplane on a grassy airfield were performed. Two different sensor systems were used in the tests: the GNSS/INS sensor and the CORREVIT optical sensor. The results showed that the effect of grass length was significant for ground velocity time course and the distances for takeoff and landing ground roll. Also, wing flaps had a measurable effect on ground roll, either by takeoff or by landing. The obtained values for the rolling resistance coefficient of the airplane’s undercarriage wheels changed along with the ground speed range.

Three different analytical models were used to calculate the ground roll of a takeoff, and the results obtained in the flight tests were compared to the calculations. All three calculation methods proved itself to be effective when tested against real measured values.

A comparison of the two sensor systems used in the tests was performed. It was concluded that both sensors were able to measure the data to determine the required parameters of the ground distance of takeoff and landing. However, both methods have drawbacks, and the intention of the authors is to seek another method. The major prerequisite is that the optimal method should be easier in handling for reliable use in multiple tests.

The data obtained in this study will be used for further analysis, and a general application of these results is to identify and verify a wheel-grass model which is a basic analytical tool of the GARFIELD information system.

## Figures and Tables

**Figure 1 sensors-19-05492-f001:**
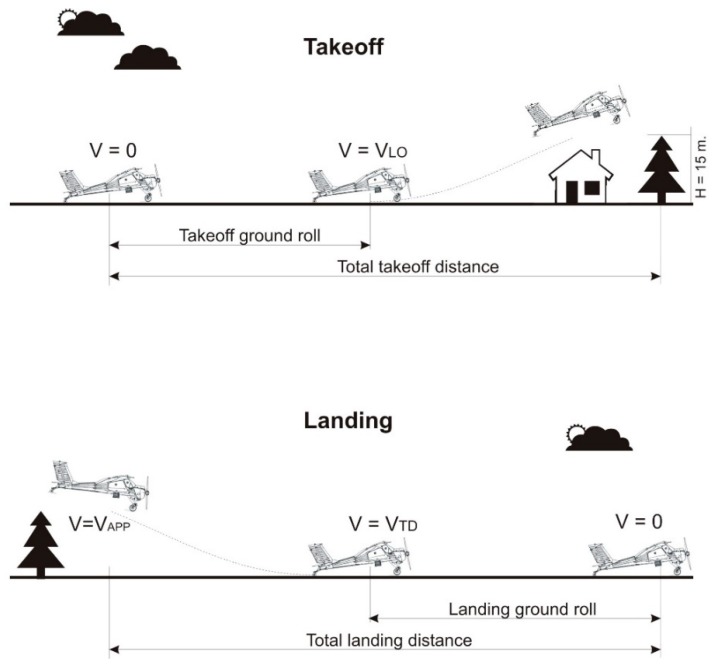
A schematic of takeoff and landing. *V_LO_*—liftoff speed, *V_APP_*—approach speed, *V_TD_*—touchdown speed.

**Figure 2 sensors-19-05492-f002:**
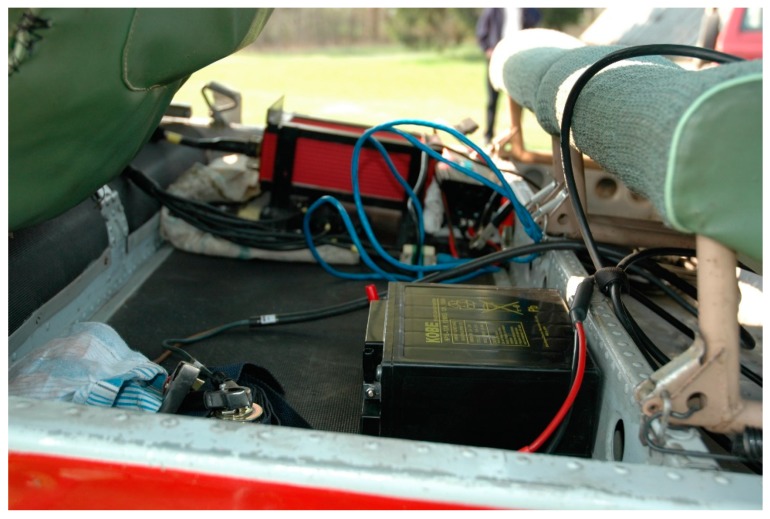
Installation of the global navigation satellite system/inertial navigation system (GNSS/INS) unit in the test airplane.

**Figure 3 sensors-19-05492-f003:**
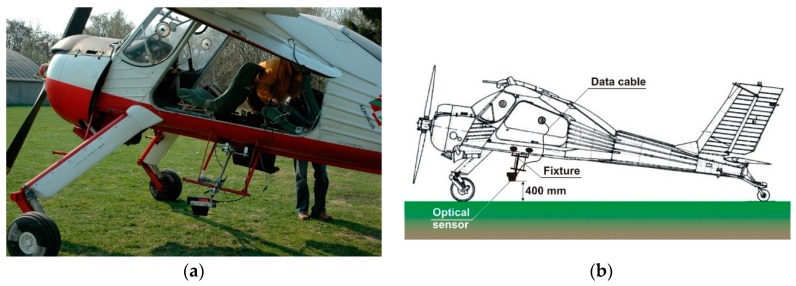
Installation of the CORREVIT L400 sensor on the test airplane. (**a**) a photograph showing the test airplane with the Correvit optical sensor, (**b**) a schematic of the installation.

**Figure 4 sensors-19-05492-f004:**
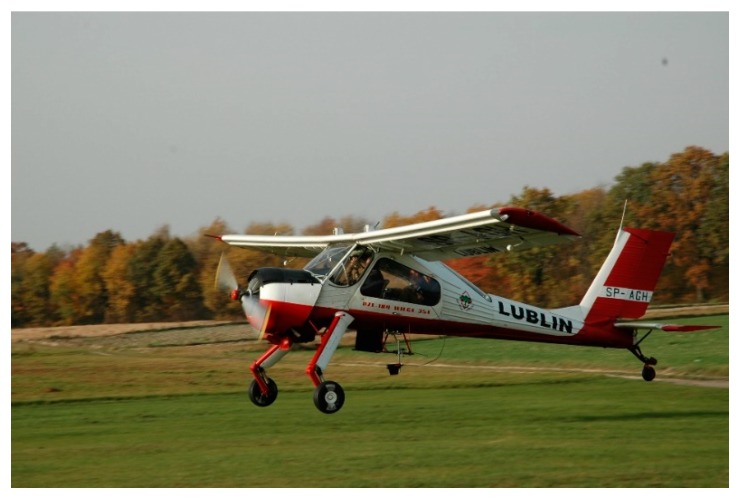
The airplane used for the flight tests: the PZL 104 Wilga with the CORREVIT optical sensor.

**Figure 5 sensors-19-05492-f005:**
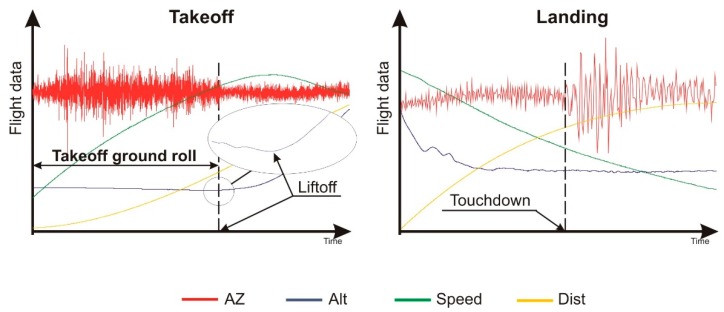
Sample results of the measurements and the idea of determining the ground roll distance.

**Figure 6 sensors-19-05492-f006:**
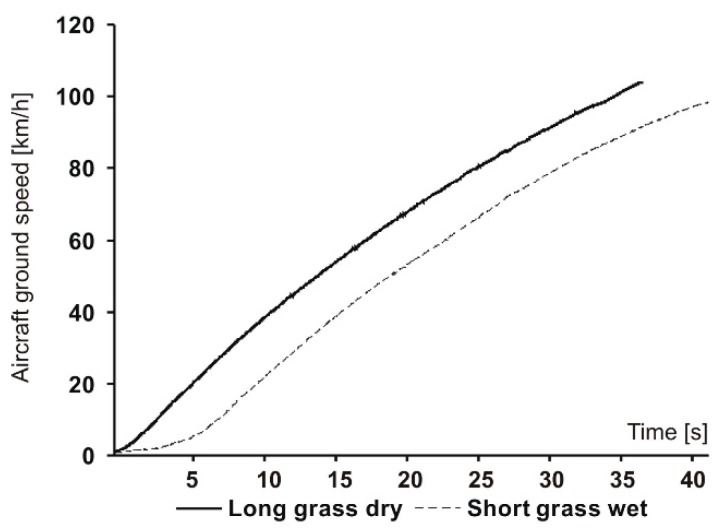
Airplane ground speed profiles for two surface conditions.

**Figure 7 sensors-19-05492-f007:**
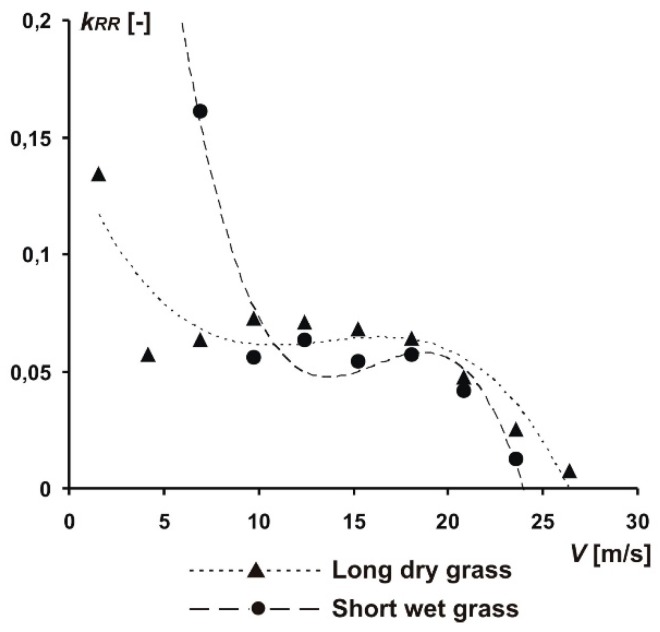
Rolling resistance coefficient for the test airplane wheels on a grassy runway.

**Figure 8 sensors-19-05492-f008:**
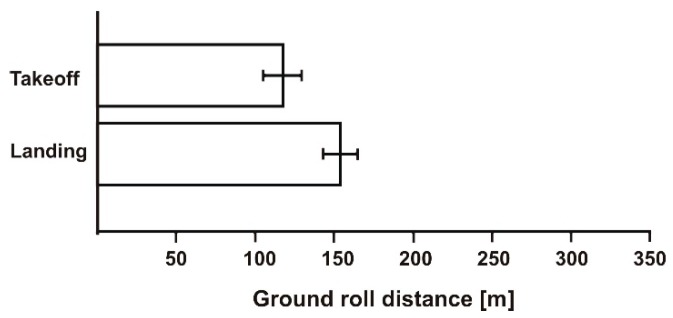
Takeoff and landing distances on short wet grass. Headwind = 5 m/s.

**Figure 9 sensors-19-05492-f009:**
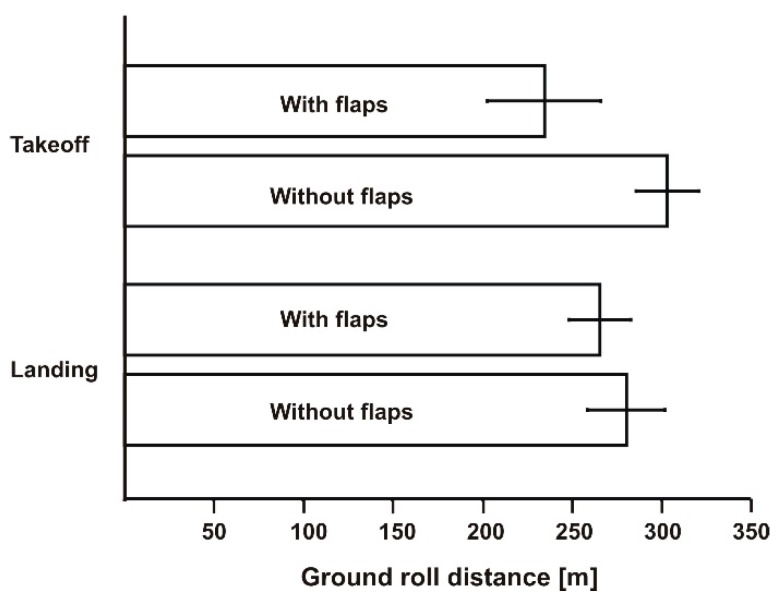
Takeoff and landing distances for the aircraft with and without flaps (wind 0–1 m/s). Long, dry grass.

**Figure 10 sensors-19-05492-f010:**
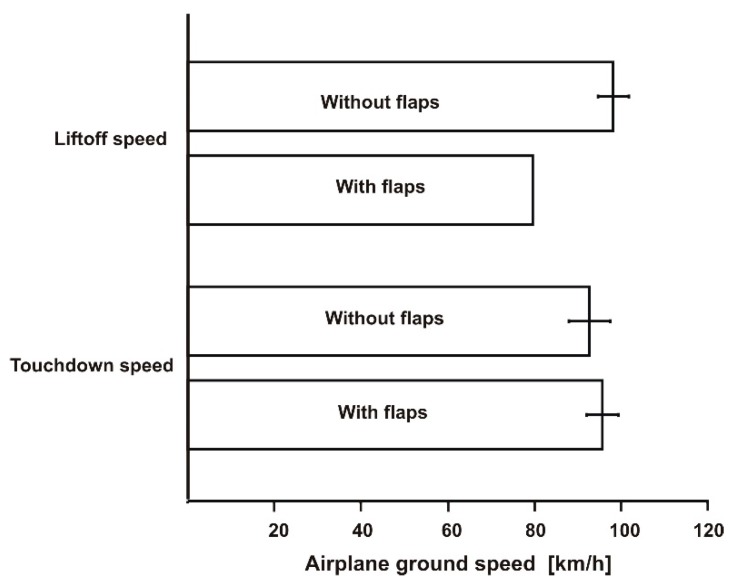
Takeoff and landing speed for the two operation modes (i.e., flaps and no flaps). Long, dry grass.

**Figure 11 sensors-19-05492-f011:**
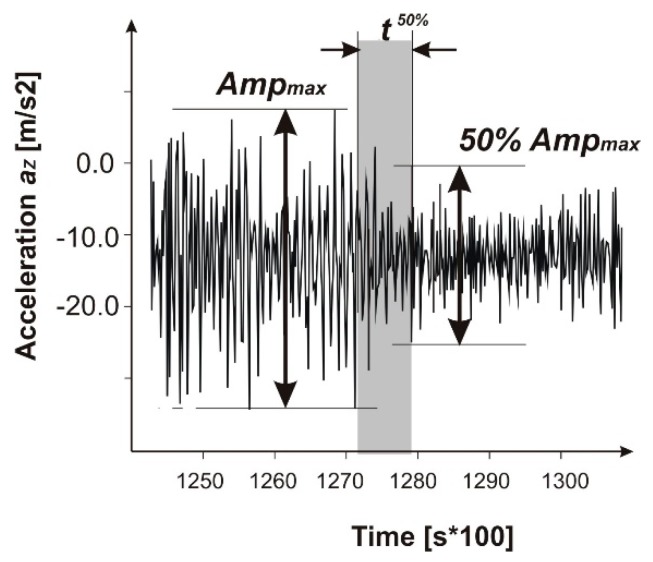
A schematic of the t^50%^ time discrimination method.

**Figure 12 sensors-19-05492-f012:**
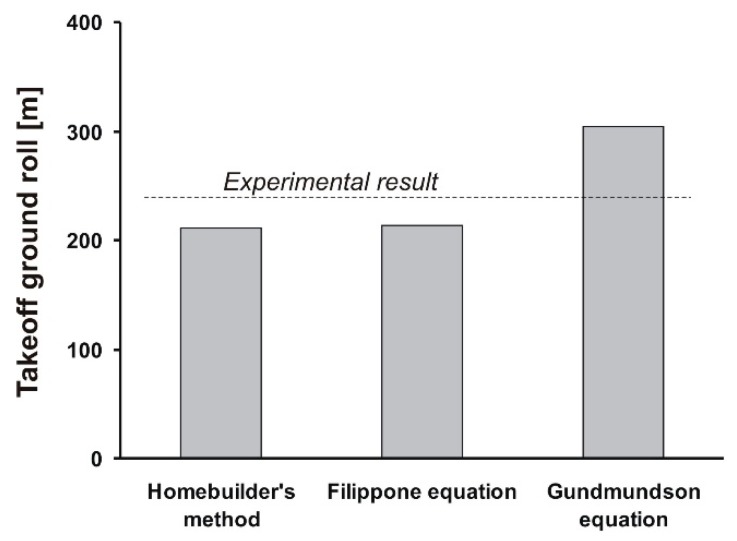
Measured versus calculated ground roll comparison.

**Table 1 sensors-19-05492-t001:** Comparison chart for the two sensors used in the study.

Functionality	Accuracy *	Installation and Operation	Data Handling **	Cost
**CORREVIT^R^ Optical Sensor**				
Full functionality in the described research	0.0027 m	Difficult and time consuming	Easy and reliable	High
**GNSS/INS Sensor**				
Full functionality in the described research	0.001 m	Easy and quick installation, but difficult data gathering	Difficult, the data can be missed	High

* The accuracy of the method depends partially on error prone human factors, (see text). The given values describe the accuracy of the measurement of basic parameters. ** Data handling for a complete measuring system, i.e., with a data acquisition device.

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
