# Peer review of "Application of GNSS/INS and an Optical Sensor for Determining Airplane Takeoff and Landing Performance on a Grassy Airfield"

_sensors, 2019, doi:10.3390/s19245492_

Round 1

Reviewer 1 Report

This is a technical paper for flight testing measurement.

(1) Firstly, I think the difficulties and meaning of this task should be emphasized. 

(2) The laser sensor is simply described. Is it designed for altitude measurement? 

(3) The taking off and landing distance have close relationship with many factors. So the distance can be determined statistically. I notice the authors give a averaged distance. However, no  more information is described.

(4)The authors are encouraged to provide more descriptions about the flight test results.

Author Response

Dear Reviewer,

thank you for your valuable comments. We've addressed all of them and have attached a file with detailed description. Also, we've made changes to the original manuscript, based on your comments and suggestions.

With best regards,

Authors

Reviewer 2 Report

The paper discusses the use of two sensors, i.e. an optical one and a GNSS/IMU integrated system, to determine some crucial parameters during the takeoff and landing of airplanes.

I don’t have the required expertise in the field of Airfield performance and therefore I cannot not evaluate the interest of the topic for the community.

From the GNSS point of view (which actually is my field of interest), a standard commercial solution is positively used. Actually what is not clear to me (and I think should be better specified), is the absolute accuracy required for the purpose of the paper. The paper demonstrated in any case that both the approaches works. What I can also suggest to the authors (maybe for future development), is to try to use the so called law-cost GNSS receivers. For very short baselines (such those adressed in the paper) the performances can be similar to those of more expensive systems (see for instance: Albéri, M., Baldoncini, M., Bottardi, C., Chiarelli, E., Fiorentini, G., Raptis, K. G. C., ... & Strati, V. (2017). Accuracy of flight altitude measured with low-cost GNSS, radar and barometer sensors: Implications for airborne radiometric surveys. Sensors, 17(8), 1889). This point should be at least adressed in the paper.

Considering Table 1 I don’t think that you can state that ”data gathering is difficult” since it is a problem or your specific hardware, not of the gnss system in general. For instance I know for sure that in aerogravimetric flights GNSS raw observations are logged up to 20Hz or even more without any missing observations.

A part from the above comments the paper is in general well written (with just some minor typos, e.g. at pag 10 line 293 “red” instead of “read” ), pictures are of good quality (maybe you can try to simplify and combine the information coming from figure 8, 9 and 10 in just one figure to facilitate the comparison).

Author Response

Dear Reviewer,

thank you for your valuable comments. We've addressed all of them and our reply is included in the attached file. Also, we've made changes to the original manuscript, based on your comments and suggestions.

With best regards,

Authors

Reviewer 3 Report

The results are relevant and I recommend this manuscript for publication in sensor.

Author Response

Dear Reviewer,

thank you for your time and effort to evaluate our manuscript. Thank you as well for the positive conclusion. 

With best regards,

Authors

Round 2

Reviewer 1 Report

There are no further comments.